# Dietary Phospholipids Enhance Growth Performance and Modulate Cold Tolerance in Meagre (*Argyrosomus regius*) Juveniles

**DOI:** 10.3390/ani11092750

**Published:** 2021-09-20

**Authors:** Ismael Hachero-Cruzado, Manuel Manchado

**Affiliations:** Istituto Andaluz de Investigación y Formación Agraria, Pesquera, Alimentaria y de la Producción Ecológica (IFAPA), Centro El Toruño, Junta de Andalucía, Camino Tiro Pichón s/n, 11500 El Puerto de Santa María, Cádiz, Spain; manuel.manchado@juntadeandalucia.es

**Keywords:** fatty acids, *Argyrosomus regius*, temperature tolerance, growth

## Abstract

**Simple Summary:**

Meagre is a target species to diversify marine aquaculture in Europe due to its high growth rates and an excellent nutritional profile. Nevertheless, this species is highly sensitive to low temperatures. The objective of this study was to evaluate the effect of dietary phospholipid (PL) levels on growth and cold tolerance. Animals fed with a PL-enriched diet grew faster and significantly reduced the risk of death and increased the lethal doses 50 and 90 without modifying the average temperature at death. Regarding lipid profiles, the cold challenge promoted a general fatty acid accumulation in the liver that was attenuated in fish fed with the PL-enriched diet preventing the negative effect of a fatty liver.

**Abstract:**

Meagre (*Argyrosomus regius*) is a fast-growing species currently produced in aquaculture. This species is highly sensitive to low environmental temperatures which results in high mortality events during production cycles. In this study, the effects of dietary phospholipids (PLs) on growth and cold tolerance were evaluated. For this purpose, control (CTRL) and PL-enriched diets (three-fold higher levels than CTRL) were supplied to meagre juveniles (12.9 ± 2.5 g) for 60 days, and growth was determined using a longitudinal approach. Weight gaining and SGR reduction were significantly different between dietary treatments. Animals fed with the PL-enriched diet were 4.1% heavier and grew 3.2% faster than those fed with the CTRL diet. Survival was higher than 98% in both groups. After finishing the growth trial, animals were submitted to two cold challenges and cold tolerance was evaluated as temperature at death (Tdeath), risk to death and lethal doses (LD) 50 and 90 using the cumulative degree cooling hours 6 h (CD6H). Tdeath ranged between 7.54 and 7.91 °C without statistical differences between dietary treatments. However, risk to death was significantly smaller (0.91-fold lower) and LD50 and LD90 were higher in animals fed with the PL-enriched than those supplied the CTRL diet. To assess the fatty acid (FA) composition of liver and brain in animals fed both diets after a cold challenge, FA profiles were determined in juveniles maintained at 14 °C and challenged at 7 °C. FA amounts increased in the liver of animals challenged at 7 °C. In contrast, several FAs reduced their levels in the PL-enriched diet with respect to CTRL indicating that these animals were able to mobilize efficiently lipids from this organ mitigating the negative effects of lipid accumulation during the cold challenge. In brain, the PL-enriched diet increased DHA level during the cold shock indicating a role in maintaining of brain functions. These results open a new research line that could improve the cold tolerance of meagre through dietary supplementation before winter.

## 1. Introduction

Water temperature is a key environmental factor that highly modulates cell physiology, metabolic rates, growth, reproduction and survival in fish [1,2]. Temperature changes modify cellular membrane fluidity, water and substrate exchanges, the enzyme catalytic activities and protein-folding [3,4]. Depending on the magnitude of these changes and the adaptative responses, animals die or suffer sublethal functional effects. Reproduction performance is highly dependent on temperature due to the capacity to modify gametogenesis, gonad maturation and the time and course of spawning. Moreover, environmental temperature acts as a modulator of embryo programming, egg quality and larvae growth trajectories [2,5,6,7,8]. As a consequence, temperature shifts have a profound impact on fish population and ecological communities and might become a serious threat to global fish biodiversity [9].

Some natural environments such as marsh ponds are highly exposed to rapid and large fluctuations of water temperature. To cope with these changes, organisms inhabiting these habitats have developed specific adaptive mechanisms such as the homeoviscous adaptation. This mechanism consists of a rapid change (in a few hours) of cell membrane lipid composition to maintain cell bilayer properties [10,11] that includes a shift in the phosphatidylcholine (PC)/phosphatidylethanolamine (PE) ratio in the relative amounts of saturated and unsaturated fatty acids (FA) and in the content of docosahexaenoic acid (DHA, 22:6n-3) [12,13,14,15,16]. In addition to these changes in cell membrane lipid composition, other processes related to lipid metabolism and lipid mobilization have been described in marine fish species subjected to low environmental temperatures. Gilthead seabream individuals cultivated at low temperatures (8–10 °C) for 20 days abnormally increase the lipid deposits in the liver to generate steatosis while the amounts of perivisceral lipids and non-polar lipid deposition in muscle are reduced [17,18,19]. This imbalanced accumulation of lipids usually results in a hepatic failure and metabolic collapse that far to be a transient condition leads to a pathogenic process that in some species is known as the “winter syndrome” [17,20].

Meagre (*Argyrosomus regius* Asso, 1801) is one of the last species identified as a target for the diversification of the marine aquaculture in Europe due to its high growth rates, good feed conversion ratios and the excellent nutritional profile accepted by consumers [21]. This species inhabits estuaries and coastal sites with temperatures ranging between 13 and 28 °C and salinities between 5‰ and 55‰ [21,22,23,24]. This species is considered a stenothermic fish species with optimal temperatures for growth between 23 and 27 °C [25]. Controlled trials have demonstrated that meagre can better adapt to warm than cold temperatures since meagre was unable to cope with a low temperature shift (from 18 to 12 °C) or temperatures below 10 °C [26]. As a consequence, high mortalities up to 100% associated with low temperatures were reported in cages located in Eastern Adriatic Sea (Bavčević, personal communication) [27] and in ponds of Southern Europe (Mazuelos, personal communication). A better understanding of factors that modulate tolerance to low temperatures is required to support meagre aquaculture.

While genetic breeding programs have proven to be a useful approach to select animals with a better tolerance to extreme temperatures [28,29,30], dietary composition also plays a key role in temperature tolerance in fish. In tilapia, replacement of coconut oil (rich in saturated FA) by fish oil rich in unsaturated FA decreases temperature at death (Tdeath) [31,32]. In the same line, cold tolerance was improved in red drum and striped bass previously fed with a diet rich in unsaturated FA [33,34]. The aim of this study is to: (1) evaluate the effects on juvenile growth of an experimental diet enriched in phospholipid (PL) with respect to a control diet (CTRL); (2) determine the effects of PL-enrichment on cold tolerance by measuring Tdeath, risk to death, lethal dose (LD) 50 and LD90 after two cold challenges; (3) establish the FA profile in liver and brain after the cold challenge as function of dietary treatments. New insights about the role of dietary PLs on growth and fish tolerance to low temperatures, as well as the associated lipid profiles are provided that are relevant for meagre aquaculture.

## 2. Materials and Methods

### 2.1. Experimental Diets

Two experimental diets, named as control (CTRL) and PL-enriched, were manufactured by Sparos S.L (Faro, Portugal). Ingredients used in the preparation of both diets are indicated in Table 1. The PL-enriched diet was added specific marine oils that resulted in 3-fold higher PL levels than the CTRL without modifying the FA profile (Table 1). Both diets were isonitrogenous (50% dry weight (dw)), isolipids (18% dw) and isocaloric (22 MJ Kg dw^−1^).

### 2.2. Fish Growth Trial

All procedures were authorized by the Bioethics and Animal Welfare Committee of IFAPA and given the registration number 26-11-15-374 by the National authorities for regulation of animal care and experimentation.

Meagre specimens (*n* = 500) were supplied by the aquaculture company Maresa (Ayamonte, Huelva, Spain). Fish were transported to IFAPA El Toruño (El Puerto de Santa Maria, Cadiz, Spain) and acclimated in 5000 L tanks for two weeks before the onset of the growth trial. Temperature, salinity and oxygen were daily monitored, and they oscillated in the ranges 18–20 °C, 32–33 ppt, and 6–8 ppm, respectively. During this period, animals were fed with a commercial dry feed (EFICO Sigma 862, Biomar, Dueñas, Spain) using belt feeders. No mortality or disease signs were observed in this period.

To evaluate the effect of both diets on growth, a total of 300 fish were randomly selected and redistributed into six-cylinder tanks (total volume 500 L; three replicates per treatment) at an initial density of 50 fish tank^−1^ (Figure 1a). To follow a longitudinal approach, all animals were anesthetized (2-phenoxyethanol, 150 ppm) and intraperitoneally injected with an electronic Passive Integrated Transponder (PIT)-tag Mini transponder (Trovan, 1.4 × 8 mm; Fish-Tags ©, Melton, UK) [35]. All fish were tagged the same day and monitored for a week before starting the trial not registering any mortality or tag losses. Average mean weight was 12.9 ± 2.5 g without statistically significant differences between tanks (size range from 6.4 to 19.4 g). During the growth trial, temperature, salinity and oxygen were daily monitored and values oscillated in the ranges 18–20 °C, 30–32 ppt, and 5–7 ppm, respectively. Diets were manually supplied during the morning (three times), while belt feeders were used in the afternoon. The amount of feed supplied was the same for all tanks (1.4% of total biomass), and it was weekly adjusted to fit the expected total biomass (based on the latest samplings). Individual weight was recorded each two weeks from the onset of the trial up to 60 days (d). Before each sampling, specimens were fasted for one day and anesthetized before handling (2-phenoxyethanol, 150 ppm). Weight and PIT-Tags were automatically registered using the FISH Reader Weight (Zeuss, Trovan, Spain). Specific growth rate (SGR; % d^−1^) = (LnW1 − LnW2)/(d1 − d2) × 100, where W1 and W2 are the body weight (g) between two consecutive samplings and d1 and d2 the range of days between samplings.

### 2.3. Cold Challenges

Once the growth trial was finished, all fish were moved to two new 5000 L tanks by diet (CTRL and PL-enriched) and fed ad libitum using belt feeders and maintaining the same diet supplied during the growth trial.

To test cold tolerance, two challenges were carried out. The first challenge (cold challenge (1) was performed 12 days and the second one (cold challenge 2) 21 days after finishing the growth trial. In each challenge, 45 fish of each diet were weighted and distributed into three tank replicates (360 L) (Figure 1b). Since all animals were intraperitoneally tagged, fish from both dietary groups were mixed in the same tanks (*n* = 15 per diet; *n* = 30 per tank). The average meagre wet weights were 77.4 ± 24.0 and 74.0 ± 21.8 g for CTRL and PL-enriched diets, respectively, without statistically significant differences between treatments (Figure 1b). It should be indicated that for both cold challenges, fish of similar size after the growth trial were selected to neutralize the possible effects associated with fish weight on cold tolerance.

All these tanks were connected to a recirculation system (RAS) equipped with a cooling, mechanical filter, skimmer, ultraviolet lights and biofilter. Animals were kept at 16 °C for 24 h and then temperature was progressively decreased by 2.7 °C per day until reaching 9 °C after 72 h. Then, water temperature decreased 0.6 °C/day and fish mortality was monitored each 6 h. Temperature was continuously recorded using a temperature data logger (HOBO PENDANTs Onset Computer Corporation, Pocasset, MA, USA). Dead fish were removed from tanks and fish tag, water temperature at death (Tdeath) and time since the beginning of the experiment was recorded. Throughout the challenges, salinity was 30 ppt, oxygen was always higher than 6 ppm and the non-ionized ammonia, nitrite and nitrate were below 0.04, 0.2 and 1.5 mg L^−1^, respectively.

To normalize data across tank replicates, the cumulative cooling degree-6hours (CD6H) parameter was used that was defined as the sum of the differences between average temperatures each 6 h (Ti) with respect to the base temperature (Tb = 10 °C for this experiment),
(1)$$CD6H=\overset{N}{\underset{i=1}{\sum }}{\left(T_{i}-T_{b}\right)}^{+}$$
where *N* is the number of time fractions (6 h) throughout challenge and the “+” superscript indicates that only positive values of the bracketed quantity are taken into account in the sum.

### 2.4. Effect of Dietary Treatments on Lipid Profile after a Cold Challenge

A new trial (23 days after finishing the growth trial) was carried out using two RAS units each of them containing triplicate tanks as indicated above. A total of 10 fish per diet (total *n* = 20) were distributed in each tank (Figure 1c). One RAS unit was set at 14 °C maintaining the water temperature constant during the trial (T14 °C). In the other RAS unit, the temperature was gradually reduced the temperature from 14 °C until reach the 7 °C (T7 °C). Temperature decreased 1.6 °C each 24 h until below 9 °C at 75 h. Thereafter, water temperature decreased 0.7 °C each 24 h until below 7 °C. After 145 h, (CD6H = 6.41), four fish of each treatment (meagre kept at 14 °C (T14 °C) or challenged at 7 °C (T7 °C)) were sacrificed with an anesthetic overdose (MS-222, >500 mg/L). Liver and brain samples were frozen in liquid nitrogen and kept at −80 °C until analysis. As no mortality was registered, the remaining non-sampled fish were moved to the stocking tanks.

Total lipid (TL) and FA methods were those previously described elsewhere [36,37]. Briefly, TL were extracted using the chloroform: methanol method and gravimetrically quantified. For FA methyl ester (FAME) determination, TL extracts were subjected to acid-catalyzed transmethylation and later they were separated and quantified by gas chromatography.

### 2.5. Statistical Analyses

Data were presented as mean ± standard deviation. All data were checked for normality, as well as for homogeneity of variances and, when necessary, log or arcsin transformation was applied. When a normal distribution and/or homogeneity of the variances were not achieved, data were analyzed using a Kolmogorov-Smirnov test. Analyses of weight gain and specific growth rate (SGR) were carried out using repeated-measures ANOVA, with dietary treatment and tank as fixed factors.

The differences in meagre Tdeath among dietary treatments were tested using a Kolmogorov-Smirnov test. For testing differences in survival and risk to death of fish exposed to cold challenge among dietary treatments, we used a binary logistic regression and a Probit analysis using dietary treatments (CTRL or PL) and fish weight as independent variables. For binary logistic regression analysis, the relationship between the dichotomous variables dead/alive, and the factors dietary treatments and fish weight at the end of cold challenge 1 (CD6H = 36) and cold challenge 2 (CD6H = 87) was tested. A Probit analysis was used to represent probability of death as a function of CD6H, and to calculate LD50 and LD90 for treatments CTRL and PL. Two-way ANOVAs were used to compare LD50 and LD90 among dietary treatments (CTRL vs. PL) and cold challenge replicates (cold challenge 1 vs. 2).

Principal Component Analyses (PCAs) were carried out to show the relationship among temperature treatments (T 14 °C and T7 °C), and diets (CTRL vs. PL) according to the FA matrices for liver and brain. Two-way ANOVAs were also used to compare liver and brain FA among diets (CTRL vs. PL) and cold challenge (T14 °C vs. T7 °C). ANOVA logistic regression and Probit analysis were performed using SPSS statistics version 22 software (IBM, Armonk, NY, USA). The multivariate analyses were carried out using the PRIMER v.6 plus PERMANOVA package (PRIMER-e, Auckland, New Zealand).

## 3. Results

### 3.1. Fish Growth Trial

Fish growth was monitored for 60 days by measuring fish weight (W) and SGR using a longitudinal approach (Figure 2). Weight increased about four-fold during the whole trial. The SGR reduced from 2.8% d^−1^ in the first fifteen days to 1.9% d^−1^ in the last two weeks of the trial. A repeated-measures ANOVA analysis identified a statistically significant interaction diet × time for weight and SGR indicating that weight gaining and SGR reduction were different between dietary treatments. Although at the beginning of the trial no differences in weight were observed, animals fed with the PL-enriched diet were on average 4.1% heavier than CTRL in the evaluation period. Differences in growth rates were more evident in the first 30 days of the trial. Overall, animals fed with the PL-enriched diet grew 3.2% faster than CTRL. Survival was not statistically different between groups, and it was higher than 98% in all cases.

### 3.2. Effect of Diets Supply on Fish Survival after Cold Shock Challenge

Average water temperature at fish death (Tdeath) ranged between 7.54 and 7.91 °C (Table 2) without statistically significant differences between dietary groups (Kolmogorov-Smirnov, *p* > 0.05). However, a logistic regression analysis between the dichotomous trait dead/alive and the factors dietary treatments and fish weight showed that PL-enriched diet reduced significantly the risk of death during cold challenges (0.91-fold lower; logistic regression, *p* < 0.05). In contrast, fish weight had no significant effect (logistic regression, *p* > 0.05). The probit survival functions for both cold challenges are depicted in Figure 3. The estimated CD6H dose to kill 50% and 90% of animals (LD50 and LD90) were in all cases 13–16% higher in fish fed with the PL-enriched than CTRL diet (Table 3; two-way ANOVA; *p* < 0.05).

### 3.3. Effect of Dietary Treatments on FA Profile after a Cold Challenge

Hepatic and brain FA profiles of fish fed both dietary treatments and kept at 14 °C (T14 °C) or challenged at 7 °C (T7 °C) were analyzed (Table 4 and Table 5). In liver, FA levels tended to be higher in fish fed with the CTRL diet than those fed with the PL-enriched diet and in fish from T7 °C than T14 °C including saturated FA (SAT), polyunsaturated FA (PUFAs), n-3 PUFAs, docosahexaenoic acid (DHA, 22:6n-3)) and arachidonic acid (ARA, 20:4n-6)) (Table 4). In contrast, the ratios monounsaturated FA (MUFA)/n-3 highly unsaturated FA (HUFA) and SAT/n-3HUFA were higher in the liver of fish fed with the PL-enriched diet than those fed the CTRL diet. No significant interaction between diet × thermal treatment was evident. In brain, only statistically significant differences in stearic acid (SA, 18:0) and ARA associated with dietary treatments were found.

Changes in hepatic and brain FA profile were further explored by a PCA multivariate analysis. In liver, the first two PCA components explained 96.1% of total variance (Figure 4). PC1 (79.2% total variance) correlated negatively with 16:0 (r = −0.531, *p* < 0.025) and 18:1n-9 (r = −0.580, *p* < 0.025) and clearly separated fish fed with PL-enriched diet and kept at T14 °C group with respect to those fed with PL-enriched diet and challenged at 7 °C (T7 °C). PC2 (17.0% total variance) correlated positively with 18:1n-9 (r = 0.696, *p* < 0.0025) and only separated samples by diets (PL vs. CTRL). In brain, PC1 and PC2 explained 96.4% of total variance (Figure 5). PC1 (72.0% total variance) positively correlated with DHA (r = 0.825, *p* < 0.0005) and separated samples by diet in fish from T7 °C group. A t-student test found significant differences for brain DHA amounts between dietary treatments at T7 °C (*t*-test; *p* < 0.05) (Figure 5b). Regarding PC2 (24.5% total variance), this axis significantly correlated with oleic acid (OA; 18:1n-9) (r = −0.826, *p* < 0.0005).

## 4. Discussion

Phospholipids are essential lipids that play a key role in somatic growth, survival, prevention of skeletal deformities and stress resistance in larvae and juveniles [38]. In this study, the effects on growth of a supplemented diet with marine oils, that resulted in three-fold higher PL amounts than the CTRL, were assessed in fast-growing meagre juveniles for 2 months. Average weight gain and growth rates obtained during the trial agreed with those data previously reported for this species [39,40,41]. Interestingly, fish fed with PL-enriched diet grew at faster growth rates and reached a higher weight than those fed with the CTRL diet. It is generally assumed that PL requirements in fish decrease as they become older since fish capacity to endogenously synthetize PLs molecules increases during development [38]. The beneficial effect of dietary PLs on growth has been associated with an enhanced transport capacity, assimilation and utilization of dietary lipids [38,42,43,44,45]. It is striking that the main effects on growth were detected during the first month of the experiment, and this effect was almost negligible at the final of the trial. It is unlikely that major ontogenic changes occur in meagre juveniles at these sizes (between 10 and 60 g) and in so small period of time (1 month) since main maturation of gut mainly occurs in the first 19 days of life in this species [46]. The most plausible hypothesis is that PLs might help the fish to cope with stressful conditions [47,48]. Prior to the beginning the experiment, fish was handled and tagged demanding more energy for acclimation that could be more effectively supplied in fish fed the PL-enriched diet.

Although both experimental diets tested in this study had a similar FA profile, major significant differences in hepatic FA levels associated with the dietary treatments were found. Meagre fed with the CTRL diet had a higher content of most FAs than those fed with the PL-enriched diet. PLs are an integral part of lipoproteins, a macromolecular complex that absorbs and transports lipids through the organism [39,42,49,50]. Thus, PLs act as a facilitator of lipid mobilization from liver to peripheric tissues by enhancing the formation of very low-density lipoproteins [42,51]. Previous studies in yellow croaker juveniles showed that high amounts of dietary PLs reduce total hepatic lipids by an increase in lipid transport and the inhibition of FA synthesis and deposition in liver [52,53]. Studies in mammals also demonstrated that PLs inhibit the synthesis and promote the oxidation of triacylglycerols in rat liver that in turn reduce the lipid contents [54,55]. Unlike the major changes found in hepatic FA contents, dietary treatments hardly provoked differences in brain (Table 3). We only detected significant differences in stearic acid (18:0) and arachidonic acid (ARA); both higher in fish fed the PL-enriched diet. Stearic acid and ARA are the main components of brain phosphatidylinositol (PI) [56]. PI and ARA have important physiological functions in brain, as regulate synaptic function through ARA metabolites FA ethanolamides anandamide (AEA) and 2-arachidonoylglycerol (2-AG) [57]; or modulate, through phosphatidylinositol 4,5-bisphosphate (PIP2), the transient receptor potential (TRP) ion channel family, curiously a group of ion channels related with the cold mediated response [58].

The estimated temperature at death (Tdeath) in both trials was below 8 °C (ranging from 7.54 and 7.91 °C). A previous study in meagre reported a loss of equilibrium (LOE) temperature ranging between 11.68 and 13.24 °C [26]. Such differences in the window of temperature sensitivity could be attributable to the reference endpoint (LOE vs. Tdeath in this study), and the cooling rate (18°C h^−1^ vs. 0.025 °C h^−1^ in this study). In any case, temperatures at the endpoint in both experiments were much higher than those reported for other fish species. So, the average LOE temperature was estimated to be 1.3 °C in goldfish and 3.4 °C in pinfish [59,60]. If we compare to marine species that cohabit with meagre in a similar distribution range, the average LOE temperature was 4.6 °C in seabass, [61] and 5 °C in gilthead sea bream [20]. These results confirm that meagre is a stenothermic species with a low thermal tolerance, far from species that share the same habitat in a similar latitude. Even so, our results provide new evidence that PLs reduced the risk of death and increased 10%–16% LD50 and LD90 after a cold challenge. There is no previous information about the effects of dietary PLs on fish tolerance to a cold challenge. However, red drum and striped bass improve cold tolerance when they are previously fed with a diet rich in unsaturated FA [33,34]. Moreover, tilapia fed with a diet in which coconut oil (rich in saturated FA) was substituted by fish oil rich in unsaturated FA decreased Tdeath improving cold tolerance [31,32].

The cold-shock challenge increased amounts of various FA levels in liver without modifying the MUFA/n-3HUFA and SAT/n-3HUFA ratios with temperature treatments indicating that hepatic lipid reserves were not involved in a homeoviscous adaptation to cold. Similarly, Ibarz, et al. [19] reported that a low temperature did not induce a homeoviscous adaptation of membranes in gilthead sea bream. However, lipids tend to accumulate quite fast in liver of this species when submitted to a low temperature [17,19,62] as observed in this study. Authors hypothesized that low temperatures generate an imbalance between lipid uptake and output in liver promoting the lipid deposition, a fast perivisceral fat mobilization, as well as a decrease in muscle lipid content and an increase in plasma FA and triglycerides levels [19]. The high deposition of FA in liver observed by these authors were responsible for dramatic modifications in the physical properties of the liver (larger, more friable and yellowish), often associated with winter disease [19]. One striking result of this study is the shift of DHA amounts in brain when fish is submitted to a cold challenge (Figure 5). DHA plays a basic role in fish adaptation to the cold, and the relative amounts of PLs rich in DHA are the major factors contributing to the maintenance of important brain functions such as signal transduction and membrane permeability during acclimatation to decreasing environmental temperature [14,63].

## 5. Conclusions

This study demonstrates that dietary PLs promote growth in meagre juveniles. The low hepatic FA levels in fish fed with the PL-enriched with respect to CTRL diet indicate that PLs enhanced the capacity of fish for lipid mobilization and energy allocation that would improve fish acclimatation and somatic growth. Moreover, PL levels reduced significantly the death risk and increased lethal doses based on CD6H to a cold challenge. One major effect of dietary PLs is the prevention of an excessive lipid accumulation in liver induced by very low temperatures. All these results joined to other biochemical changes in brain (e.g., increase of DHA and ARA content in brain), indicate that a diet enriched in PLs could be a useful prevention measure to improve fish welfare and reduce mortalities during winter and it should be carefully taken into consideration for diet formulation in this fast-growing species.

## Figures and Tables

**Figure 1 animals-11-02750-f001:**
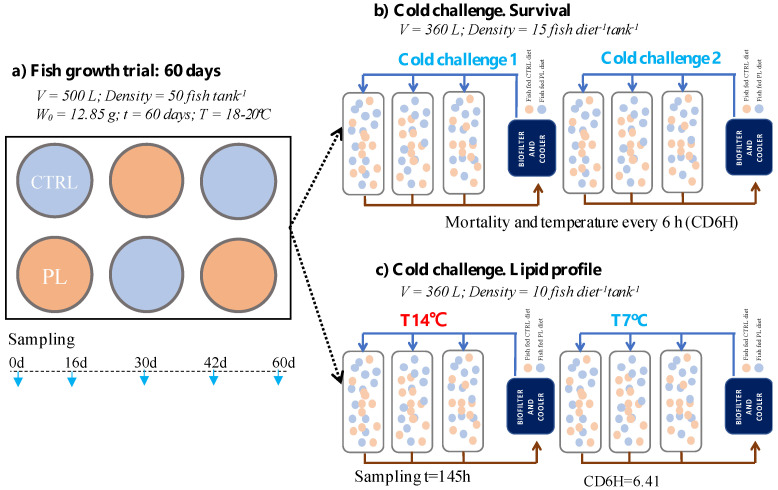
Overview of experimental design. (**a**) Scheme of the fish growth trial indicating the tanks fed with the control (CTRL, blue) or PL-enriched (PL, orange) diets. Average weight for fish, tank volume, fish densities and sampling days are shown; (**b**) scheme of cold challenge designs for survival analysis. Two challenges (1 and 2) were carried out. Tank volume and fish densities are shown. Mortality and temperature were recorded every 6 h to estimate the cumulative cooling degree-6hours (CD6H); (**c**) scheme of cold challenge to establish lipid profile in brain and liver. Sampling time, CD6H for T7 °C treatment, tank volume and fish densities are shown.

**Figure 2 animals-11-02750-f002:**
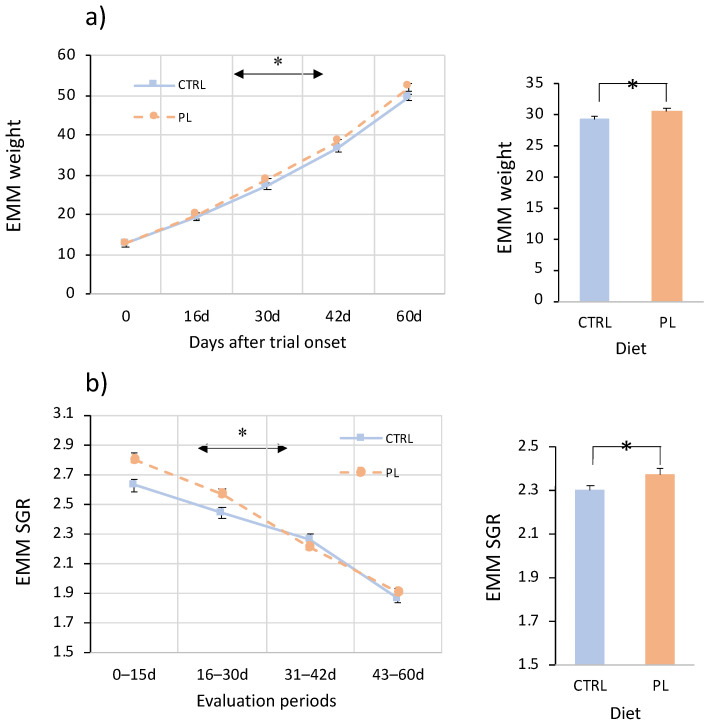
Weight (**a**) and specific growth rates (SGR) (**b**) of animals fed with the control (CTRL, blue) or PL-enriched (PL, orange) diets. Lines represent the estimated marginal means (EMM) after a repeated measures ANOVA using the weight at different samplings or the SGR in the evaluation periods. Asterisks on the horizontal denote if within-subjects for interaction diet × time are statistically significant. Bars on the right represent the EMM for weight and SGR in the whole period. Asterisks denote if between-subjects are statistically significant.

**Figure 3 animals-11-02750-f003:**
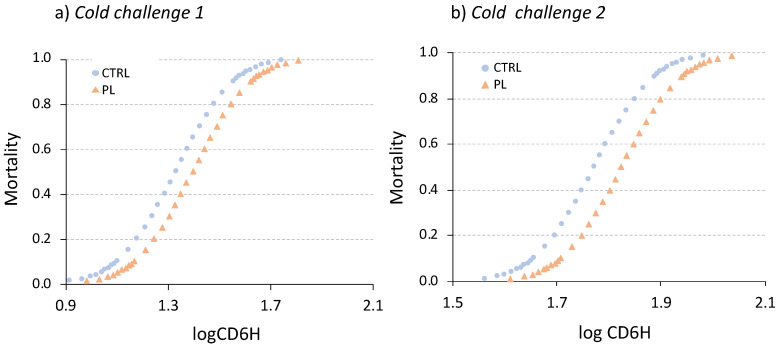
Mortality curves fitted by probit analysis as function of cumulative cooling degree-6 h (CD6H) for each dietary treatment groups. Data for each cold challenge (named as 1 (**a**) and 2 (**b**)) are indicated. Animals fed with the control diet (CTRL) are in blue circles and those fed with PL-enriched diet in orange triangles.

**Figure 4 animals-11-02750-f004:**
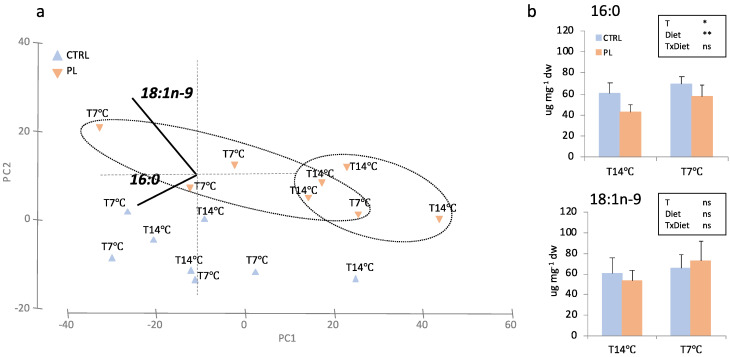
Hepatic FA amounts in animals fed with the control (CTRL, blue) or PL-enriched (PL, orange) diets. (**a**) Principal component analysis (PCA) plot based on FA composition (in μg FA mg dw^−1^); (**b**) hepatic content of FA significantly correlated with PC1: 16:0 and 18:1n-9; and PC2: 18:1n-9. Data were expressed as mean ± SD (*n* = 4). Results of two-way ANOVA are presented in the square (T = temperature; diet = dietary treatment). Asterisk above bars indicates significant differences between temperature treatments (*t*-test; *p* < 0.05).

**Figure 5 animals-11-02750-f005:**
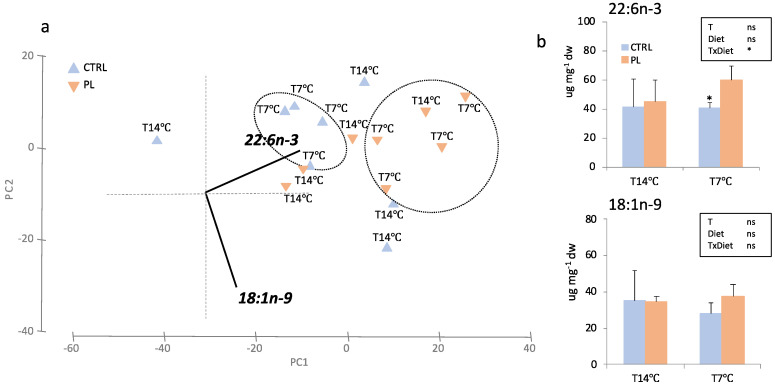
FA amounts in brain of animals fed with the control (CTRL, blue) or PL-enriched (PL, orange) diets. (**a**) Principal component analysis (PCA) plot based on FA composition (in μg FA mg dw^−1^). (**b**) Brain content of FA significantly correlated with PC1: DHA; and PC2: 18:1n-9. Data were expressed as mean ± SD (*n* = 4). Results of two-way ANOVA are presented in the square (T = temperature; diet = dietary treatment). Asterisk above bars indicates significant differences between temperature treatments (*t*-test; *p* < 0.05). * *p* < 0.05; ** *p* < 0.01; *** *p* < 0.001.

**Table 1 animals-11-02750-t001:** Formulation and proximate composition (% dw) of experimental control (CTRL) and phospholipid(PL)-enriched diets.

Dietary Composition	Diets	
	CTRL	PL-Enriched
Ingredients		
Fish meal	17.5	17.5
Squid meal	2.5	2.5
Soybean meal	5	5
Whole wheat	3.04	3.04
Soy protein concentrate	8	8
Pea protein	4.5	4.5
Wheat gluten	14	14
Corn gluten	10	10
Soluble fish protein concentrate (CPSP^®^ 90)	2.5	2.5
Amino acids enrichment	1.8	1.8
Fish oil	14.5	8.5
Rapeseed oil	0	1.5
Tuna oil	0	1
Krill oil	0	3.5
Soy lecithin	0.5	0.5
Aquatex G2000^®^	9	9
Premix^®^	1	1
Lutavit C35^®^	0.43	0.43
Lutavit E50^®^	0.03	0.03
Monocalcium phosphate	2.5	2.5
Sel-Plex^®^	0.09	0.09
Antioxidant	0.2	0.2
Glycerol	2.5	2.5
Binder	0.5	0.5
Proximate composition		
Protein	50.2	50.2
Lipid	18.3	18.4
Starch	7.6	7.6
Ash	4.9	4.9
Lipid profile		
Total PL	0.94	2.57
Saturated Fatty acids (FA)	4.2	4.5
Monounsaturated FA	7.3	6.4
Polyunsaturated FA	4.6	4.9

**Table 2 animals-11-02750-t002:** Wet weight (g) and temperature at death (Tdeath) of meagre fed with the CTRL and PL-enriched diets. Data for the two cold challenges (named as 1 and 2) are indicated.

Trial	Diet	Weight (g)	95% Confidence Limit		Tdeath	95% Confidence Limit	
			Lower	Upper		Lower	Upper
Cold Challenge 1	CTRL	69.71	60.48	78.96	7.75	7.59	7.91
	PL	63.45	54.15	72.76	7.66	7.54	7.78
Cold Challenge 2	CTRL	81.20	75.21	87.16	7.80	7.76	7.84
	PL	79.10	74.33	83.88	7.79	7.75	7.83

Values are expressed as mean and confidence limit 95%; cold challenge 1, *n* = 36; cold challenge 2, *n* = 38.

**Table 3 animals-11-02750-t003:** Lethal doses (LD50 and LD90) for CD6H data fitted to probit-log(dose) regression models in meagre fed with CTRL and PL-enriched diets. Data for the two cold challenges (named as 1 and 2) are indicated.

Experiment	Diet	LD50 (CD6H)	95% Confidence Limit		LD90 (CD6H)	95% Confidence Limit	
			Lower	Upper		Lower	Upper
Cold Challenge 1	CTRL	21.51	18.54	23.84	36.30	33.27	40.55
	PL	24.93	21.18	27.92	42.07	38.29	47.16
Cold Challenge 2	CTRL	59.18	57.57	60.72	77.37	75.27	79.75
	PL	66.61	64.95	68.26	87.09	84.54	90.05

Values are expressed as mean and confidence limit 95%; cold challenge 1, *n* = 36; cold challenge 2, *n* = 38.

**Table 4 animals-11-02750-t004:** FA composition (µg mg dw^−1^) in liver of meagre fed with CTRL and PL-enriched diets and kept at 14 °C (T14 °C) or challenged at 7 °C (T7 °C) (mean ± SD, *n* = 4).

Fatty Acid	CTRL		PL		P(diet)	P(T14/7)	P(d * T14/7)
	T14 °C	T7 °C	T14 °C	T7 °C			
14:0	12.8 ± 2.1	13.9 ± 1.0	8.8 ± 1.3	11.8 ± 2.0	0.003	0.026	0.270
15:0	1.8 ± 0.3	1.9 ± 0.2	1.0 ± 0.1	1.6 ± 0.3	0.001	0.011	0.141
16:0	60.2 ± 10.1	69.5 ± 6.9	43.2 ± 6.9	58.0 ± 11.0	0.008	0.019	0.555
16:1n-7	29.2 ± 5.0	32.2 ± 2.4	16.5 ± 1.8	23.3 ± 4.1	0.000	0.018	0.301
17:0	1.5 ± 0.5	2.3 ± 0.1	3.7 ± 0.3	5.2 ± 0.8	0.000	0.001	0.226
18:0	9.5 ± 3.4	11.2 ± 1.2	8.8 ± 2.2	10.9 ± 5.68	0.779	0.311	0.908
18:1n-9	60.3 ± 15.1	66.1 ± 12.2	53.2 ± 10.3	72.9 ± 19.6	0.985	0.110	0.361
18:2n-6	41.1 ± 7.1	44.7 ± 7.3	30.4 ± 4.8	43.8 ± 8.6	0.134	0.034	0.192
18:3n-6	0.5 ± 0.1	0.5 ± 0.1	0.2 ± 0.1	0.4 ± 0.1	0.006	0.055	0.179
18:3n-3	5.6 ± 0.8	6.0 ± 0.9	4.6 ± 0.8	6.6 ± 1.37	0.666	0.029	0.136
20:1n-9	24.5 ± 4.2	28.1 ± 2.4	14.1 ± 2.6	17.3 ± 5.8	0.000	0.114	0.933
20:2n-6	1.7 ± 0.2	1.9 ± 0.4	1.1 ± 0.1	1.6 ± 0.3	0.011	0.033	0.532
20:3n-6	0.4 ± 0.2	0.4 ± 0.2	0.2 ± 0.1	0.3 ± 0.2	0.142	0.579	0.934
20:4n-6 (ARA)	1.5 ± 0.3	1.6 ± 0.2	0.9 ± 0.1	1.3 ± 0.1	0.002	0.037	0.277
20:3n-3	1.1 ± 0.2	1.0 ± 0.2	0.8 ± 0.2	0.9 ± 0.2	0.117	0.853	0.292
20:4n-3	0.5 ± 0.1	0.5 ± 0.1	0.4 ± 0.0	0.5 ± 0.1	0.303	0.250	0.255
20:5n-3 (EPA)	17.7 ± 2.0	18.7 ± 3.0	10.7 ± 2.4	14.6 ± 2.8	0.001	0.081	0.276
22:1n-9	3.5 ± 0.5	4.0 ± 0.5	2.3 ± 0.3	3.0 ± 0.8	0.001	0.052	0.822
22:6n-3 (DHA)	33.5 ± 4.1	34.8 ± 5.2	17.9 ± 3.1	25.7 ± 3.4	0.000	0.046	0.137
SAT	85.8 ± 14.7	98.8 ± 8.2	65.7 ± 10.4	87.5 ± 19.2	0.042	0.026	0.538
MUFA	119.3 ± 23.6	132.4 ± 16.3	87.5 ± 14.8	118.0 ± 30.0	0.058	0.071	0.449
PUFA	103.4 ± 14.7	110.1 ± 14.5	67.4 ± 10.1	95.7 ± 16.1	0.004	0.028	0.148
n-3 PUFA	58.3 ± 7.0	60.9 ± 8.9	34.4 ± 6.3	48.3 ± 7.5	0.000	0.049	0.159
n-6 PUFA	44.6 ± 7.7	48.6 ± 8.1	32.8 ± 5.1	47.0 ± 9.1	0.104	0.034	0.204
n-3 HUFA	52.7 ± 6.3	54.9 ± 8.3	29.8 ± 5.7	41.7 ± 6.3	0.000	0.059	0.176
DHA/EPA	1.89 ± 0.1	1.9 ± 0.1	1.7 ± 0.1	1.8 ± 0.1	0.026	0.644	0.385
ARA/EPA	0.1 ± 0.0	0.1 ± 0.0	0.1 ± 0.0	0.1 ± 0.0	0.581	0.715	0.831
MUFA/SAT	1.4 ± 0.1	1.3 ± 0.1	1.3 ± 0.1	1.3 ± 0.1	0.566	0.639	0.516
MUFA/PUFA	1.1 ± 0.1	1.2 ± 0.1	1.3 ± 0.1	1.2 ± 0.1	0.135	0.881	0.220
MUFA/n-3PUFA	2.0 ± 0.2	2.2 ± 0.2	2.6 ± 0.4	2.4 ± 0.4	0.031	0.961	0.358
MUFA/n-3HUFA	2.2 ± 0.2	2.4 ± 0.3	3.0 ± 0.5	2.8 ± 0.5	0.016	0.943	0.398
SAT/PUFA	0.8 ± 0.0	0.9 ± 0.1	1.0 ± 0.1	0.9 ± 0.1	0.036	0.832	0.052
SAT/n-3PUFA	1.5 ± 0.1	1.6 ± 0.2	1.9 ± 0.2	1.8 ± 0.2	0.004	0.741	0.117
SAT/n-3HUFA	1.6 ± 0.1	1.8 ± 0.20	2.2 ± 0.2	2.1 ± 0.3	0.002	0.744	0.152

Results for statistical analysis after two-way ANOVA analysis using diet (d) and thermal treatments (T14 °C/ T7 °C; T14/7) as fixed factors are shown. SAT, saturated FA; MUFA, monounsaturated FA; PUFA, polyunsaturated FA; HUFA, highly unsaturated FA; DHA, docosahexaenoic acid; EPA, eicosapentaenoic acid; ARA, arachidonic acid.

**Table 5 animals-11-02750-t005:** FA composition (in µg mgdw^−1^) in brain of meagre fed experimental diets CTRL and PL and kept at 14 °C (T14 °C) or challenged at 7 °C (T7 °C) (mean ± SD, *n* = 4).

Fatty Acid	CTRL		PL		P(diet)	P(T14/7)	P(d * T14/7)
	T14 °C	T7 °C	T14 °C	T7 °C			
14:0	2.6 ± 1.0	1.0 ± 0.3	1.3 ± 0.6	1.8 ± 0.5	0.547	0.145	0.009
15:0	nd	nd	nd	nd	-	-	-
16:0	30.8 ± 10.6	25.2 ± 2.1	30.6 ± 4.4	35.9 ± 4.4	0.119	0.960	0.106
16:1n-7	6.0 ± 3.3	3.1 ± 0.6	3.7 ± 0.7	4.0 ± 0.8	0.438	0.159	0.097
17:0	nd	nd	nd	nd	0.994	0.816	0.342
18:0	15.7 ± 7.5	16.1 ± 1.6	19.5 ± 3.0	23.4 ± 2.7	0.025	0.721	0.430
18:1n-9	35.1 ± 16.5	28.1 ± 5.8	34.4 ± 3.0	37.5 ± 6.5	0.376	0.683	0.310
18:2n-6	9.5 ± 5.4	4.2 ± 0.9	5.4 ± 1.5	6.3 ± 0.8	0.500	0.149	0.053
18:3n-6	nd	nd	nd	nd	-	-	-
18:3n-3	1.6 ± 1.1	0.4 ± 0.1	0.6 ± 0.3	0.8 ± 0.1	0.372	0.141	0.032
20:0	nd	nd	nd	nd			
20:1n-9	4.9 ± 2.4	2.3 ± 0.3	2.5 ± 0.8	2.9 ± 0.5	0.205	0.121	0.041
20:2n-6	0.7 ± 0.2	0.4 ± 0.1	0.4 ± 0.2	0.5 ± 0.2	0.153	0.381	0.052
20:3n-6	nd	nd	nd	nd	-	-	-
20:4n-6 (ARA)	2.2 ± 0.8	2.3 ± 0.3	2.7 ± 0.6	3.2 ± 0.2	0.024	0.318	0.420
20:3n-3	nd	nd	nd	nd	-	-	-
20:4n-3	nd	nd	nd	nd	-	-	-
20:5n-3 (EPA)	8.3 ± 3.6	6.2 ± 0.7	8.1 ± 0.9	9.5 ± 2.0	0.169	0.731	0.126
22:1n-9	1.2 ± 0.8	0.8 ± 0.2	1.0 ± 0.2	0.9 ± 0.2	0.764	0.283	0.359
22:6n-3 (DHA)	41.4 ± 18.9	40.8 ± 3.0	45.0 ± 14.4	60.0 ± 9.7	0.103	0.287	0.025
SAT	49.1 ± 18.0	42.3 ± 3.5	51.4 ± 7.3	61.1 ± 7.5	0.068	0.782	0.143
MUFA	47.2 ± 22.7	34.3 ± 6.8	41.6 ± 3.4	45.3 ± 7.6	0.675	0.474	0.210
PUFA	63.8 ± 24.3	54.4 ± 3.7	62.3 ± 14.4	80.4 ± 8.6	0.125	0.570	0.090
n-3 PUFA	51.3 ± 20.7	47.4 ± 3.2	53.8 ± 14.3	70.4 ± 8.1	0.080	0.357	0.150
n-6 PUFA	12.5 ± 5.9	7.0 ± 1.3	8.6 ± 1.2	10.1 ± 0.6	0.781	0.218	0.043
n-3 HUFA	49.7 ± 20.6	47.0 ± 3.2	53.2 ± 14.3	69.5 ± 8.1	0.074	0.323	0.178
DHA/EPA	5.4 ± 3.5	6.6 ± 0.9	5.6 ± 2.0	6.6 ± 2.2	0.940	0.355	0.927
ARA/EPA	0.3 ± 0.1	0.4 ± 0.0	0.3 ± 0.1	0.3 ± 0.1	0.801	0.334	0.475
MUFA/SAT	1.0 ± 0.3	0.8 ± 0.1	0.8 ± 0.2	0.7 ± 0.2	0.330	0.243	0.604
MUFA/PUFA	0.8 ± 0.3	0.6 ± 0.1	0.7 ± 0.2	0.6 ± 0.1	0.524	0.179	0.975
MUFA/n-3PUFA	1.0 ± 0.4	0.7 ± 0.2	0.8 ± 0.3	0.6 ± 0.2	0.378	0.121	0.712
MUFA/n-3HUFA	1.0 ± 0.4	0.7 ± 0.2	0.8 ± 0.3	0.7 ± 0.2	0.344	0.110	0.627
SAT/PUFA	0.8 ± 0.0	0.8 ± 0.1	0.8 ± 0.1	0.8 ± 0.0	0.459	0.234	0.176
SAT/n-3PUFA	1.0 ± 0.1	0.9 ± 0.1	1.0 ± 0.1	0.9 ± 0.0	0.776	0.083	0.829
SAT/n-3HUFA	1.0 ± 0.1	0.9 ± 0.1	1.0 ± 0.2	0.9 ± 0.0	0.634	0.130	0.929

Results for statistical analysis after two-way ANOVA analysis using diet and temperature treatment (T14 °C/ T7 °C; T14/7) as fixed factors are shown. nd, not detected; SAT, saturated FA; MUFA, monounsaturated FA; PUFA, polyunsaturated FA; HUFA, highly unsaturated FA; DHA, docosahexaenoic acid; EPA, eicosapentaenoic acid; ARA, arachidonic acid.

## Data Availability

Data will be made available upon any reasonable request to the corresponding author.

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
