# Peer review of "Dietary Phospholipids Enhance Growth Performance and Modulate Cold Tolerance in Meagre (Argyrosomus regius) Juveniles"

_animals, 2021, doi:10.3390/ani11092750_

Round 1
Reviewer 1 Report
The authors investigated the effect of dietary phospholipids on growth performance and mitigating the cold tolerance in meagre (Argyrosomus regius) juveniles. They designed two treatments to test the fish response to temperature. This manuscript (MS) was clearly written and easy to understand. Although they covered a wide range of parameters, still some gaps are here in terms of provided data. This work could help the sustainability of this species farming if this work could be designed better and more parameters measured. Therefore, some major issues significantly compromised the quality of this MS.
Major comments:
- The experimental design of this MS can be questioned as a temperature too far from “optimum” for this fish species was selected as a control. Preferred temperature range estimated for juvenile meagre between 26 and 30 °C. In your control, the temperature is 14 degrees for fatty acids analysis. As we know, temperature hugely affects lipid metabolism, and the provided data is a bit misleading and cannot support the conclusion.
- They did not provide adequate data and evidence for supporting their hypothesis. The data of growth, mortality and fatty acids were repeated in both Tables and Figures. For publishing in the Q1 journal, enough data from an appropriate experimental design should be provided.
- I did not go through to give minor comments as, regretfully, my decision was Reject.
Best regards
Author Response
Response to REVIEWER1
The authors investigated the effect of dietary phospholipids on growth performance and mitigating the cold tolerance in meagre (Argyrosomus regius) juveniles. They designed two treatments to test the fish response to temperature. This manuscript (MS) was clearly written and easy to understand. Although they covered a wide range of parameters, still some gaps are here in terms of provided data. This work could help the sustainability of this species farming if this work could be designed better and more parameters measured. Therefore, some major issues significantly compromised the quality of this MS.
Major comments:
The experimental design of this MS can be questioned as a temperature too far from “optimum” for this fish species was selected as a control. Preferred temperature range estimated for juvenile meagre between 26 and 30 °C. In your control, the temperature is 14 degrees for fatty acids analysis. As we know, temperature hugely affects lipid metabolism, and the provided data is a bit misleading and cannot support the conclusion.
Thanks to the reviewer for his/her comment but we do not fully agree with it. The reviewer should consider that this study was structured in three parts: a) Evaluation of growth using standard production conditions. This trial was carried out at temperatures ranging between 18 and 20ºC that are routinely used in farms across the Mediterranean and Atlantic coasts. Moreover, this range of temperatures is the most frequent used in the hatcheries; b) Evaluation of cold tolerance. This trait was evaluated by two cold challenges in which a low temperature within the physiological range (14-16ºC) was set. At this range, no symptoms or behavioural alterations were visualized; c) Evaluation of the cold challenge on fatty acid composition using as reference a temperature (14ºC) within the physiological range that does not modify swimming behaviour o produces mortality. The reviewer refers to an optimal temperature range calculated based on oxygen consumption rate (OCR) (Kir et al 2017) that is also related to the range of optimal growth temperatures. However, in this study we are more interested in determine what are the effects of very low temperatures that produce mortality (7ºC) compared to the low temperatures within the physiological (14ºC). This comparison would allow to identify the unsuccessful compensatory and adaptive mechanism triggered by meagre to adapt to very low temperatures. From this perspective, we think that temperature selected for control in this experiment is more suitable than proposed by the reviewer to support our conclusions.
Reference
Kır, M.; Sunar, M.C.; Altındağ, B.C. Thermal tolerance and preferred temperature range of juvenile meagre acclimated to four temperatures. Journal of Thermal Biology 2017, 65, 125-129, doi:https://doi.org/10.1016/j.jtherbio.2017.02.018.
They did not provide adequate data and evidence for supporting their hypothesis.
The data of growth, mortality and fatty acids were repeated in both Tables and Figures. For publishing in the Q1 journal, enough data from an appropriate experimental design should be provided.
We think that reviewer confused the data provided in the different tables and figures. We did four trials, one to evaluate growth, two cold challenges to determine cold tolerance, and a third cold challenge to determine the effects on FA composition. No data were repeated. Figures represent data obtained for the evaluation of growth. Tables 2 and 3 indicate the weight and survival of meagre juveniles used for cold challenges. Tables 4 and 5 are the FA profiles in liver and brain. We consider that we performed enough experiments (four trials) to support our data giving robustness to our data.
Reviewer 2 Report
The MS is well planned, executed and written. It deals with a very important issue, which is thermal tolerance and the search for methods to increase resistance to cold in thermophilic fish, especially those with a narrow range of optimal temperatures. In my opinion, however, the work requires improvement, supplementation or clarification in several places. My detailed comments are included in the text (MS) in the form of comments.

Author Response
Reviewer 2
The MS is well planned, executed and written. It deals with a very important issue, which is thermal tolerance and the search for methods to increase resistance to cold in thermophilic fish, especially those with a narrow range of optimal temperatures. In my opinion, however, the work requires improvement, supplementation or clarification in several places. My detailed comments are included in the text (MS) in the form of comments.
Thanks to the reviewer for his/her positive comments on the MS.
Line 42. Fish are cold-blooded animals and their temperature depends (in most cases) on the water temperature. Temperature affects all forms of fish activity (from spawning to growth). Maybe you could add a bit of such information here, because the journal Animals publishes not only in the field of aquaculture and ichthyology.
We have included a paragraph at the beginning of the introduction section (lines 43-53) describing more aspects related to the effects of temperatures on reproduction, larval development and growth.
Line 63. I suggest to add paragraph about influence of lethal/sub-lethal temperature on fish. Use e.g. Journal of Thermal Biology, 2014, 45, pp. 62–68; Czech Journal of Animal Science, 2011, 56(2), pp. 70–80; Polskie Archiwum Hydrobiologii, 1997, 44(1-2), pp. 139–143;
We have included some information about sublethal effects of temperature citing the references indicate above (lines 46-51).
Line 65. Also, please add information that not only temperature level might be lethal. improper for fish but also their changes and fluctuations.
We have added some information about the effects of temperature fluctuations in line 77-78.
Line 74. delete (
This was corrected
Line 102. water parameteres: T, DO, ammonia, salinity, photoperiod, ect ..... should be present here
The parameters were included as requested (line 115-116).
Line 107. in which? add dose?
The anesthetic and dose were included (line 122).
Line 108. Each??
Tank volume was added in text
Line 117. what was the expected growth rate?
This was explained in line 131-132
Line 120. Do you mean 2-phenoxyethanol?
Yes. It has been corrected (line 134)
Figure 1. why is the difference in densities?
The densities changed according to the objectives of each trial and the number of animals required to achieve robust data.
In the growth trial, we used 50 fish per tank that could be similar to densities in the hatcheries that could provide the number of animals required for the challenge experiments. In the cold challenge we used 15 fish per diet in each tank to precisely quantify the survival and mortalities. In the last challenge to determine the FA composition, only 4 animals per diet were going to be analyzed, so, we reduced fish number to 10 per diet in the tanks to get the samples in similar conditions while reducing stress for unnecessary animals.
Line 140. Is any statistical differences? Figure 2. see lines 141-142 - the fish from control groups were heavier ...
No. For both cold challenges, fish of similar size after the growth trial were selected to neutralize effects associated with fish weight on cold tolerance.
It is clarified in the text (lines 154-158).
Line 151. Salinity
It was added (Lines 166-167).
Table 4 and 5. marked ARA, EPA and DHA
Done.
Line 292. Please, start the discussion from whole word, not abbreviation
Done.
Reviewer 3 Report
In the manuscript submitted by I. Hachero-Cruzado and M. Manchado, the authors studied the effects of dietary supplementation with phospholipids on the growth and cold tolerance of meagre fish. Using a nice experimental design, the authors performed a growth trial with two different diets (60 days), which was followed by two cold challenges to evaluating the cold tolerance of the fish fed with the different diets. The results are interesting and provide new evidence that could contribute to improving the aquaculture production of meagre species, on the one hand by optimizing its somatic growth, and on the other, by providing new information that can help to improve the cold tolerance of this species to avoid the high mortalities associated with changes in temperature. In addition, the obtained results could be extended to other species that share the same problem with low temperatures.
In my opinion, the work is worthy of being published in Animals. However, authors should review the following minor points before approval:
-Line 19: the word “occur” can be removed.
-Lines 14 and 29: LD50 and LD90 were higher, or lower? Please check.
-Line 45: Fish inhabiting such habitats have developed
-Figure 1a: Please indicate in the figure the temperature (18-20) at which the fish were kept during the growth trial.
-Line 100 and bellow: The authors mentioned that 500 fish were used. However, only 300 fish were used for both the Fish growth trial and the cold challenges. Please check.
-Line 102 and Figure1a: 500L or 5,000L?
-Lines 134 and 135: please review this sentence. It is not very clear whether all the fish were pooled, or one of the two tanks was dedicated to each diet.
-Figure 2: please check the colours for the identification of the groups. The figure legend and figure caption do not match. In addition, in order to be consistent with Figure 1, the CTRL group should be presented in blue.
-Line 300: capacity... increases
-Lines 298-302: The authors mentioned that the requirements of PL in fish decrease with their development due to an increase in the capacity to endogenously synthesize PL with ageing. Said that the results show that fish fed the PL diet presented increased SGR in the first half of the growth trial, but not in the second part (Figure 2B). It might be thought, therefore, that the growth-promoting effects of PL supplementation will not be found in older animals. The authors should discuss this a bit deeper this section, and revise the use of “however” in line 301, as the observed results do not contradict the classical assumption about PL requirements.
-Lines 326-327: Considering that PL group grew more than CTRL animals, it is surprising that, as showed in Table 2, the wet weight at the death of PL-diet fed fish was lower than of the CTRL group. The authors neither mention nor discuss these results. Were the death fish of the PL group outliers for the body weight and SGR?
-Line 363: Considering the presented results, that conclusion statement is only accurate for meagre under 30-35 grams (the first part of the growth trial). Please review it or indicate the limitations of the study and its derived conclusions.
Author Response
Reviewer 3
In the manuscript submitted by I. Hachero-Cruzado and M. Manchado, the authors studied the effects of dietary supplementation with phospholipids on the growth and cold tolerance of meagre fish. Using a nice experimental design, the authors performed a growth trial with two different diets (60 days), which was followed by two cold challenges to evaluating the cold tolerance of the fish fed with the different diets. The results are interesting and provide new evidence that could contribute to improving the aquaculture production of meagre species, on the one hand by optimizing its somatic growth, and on the other, by providing new information that can help to improve the cold tolerance of this species to avoid the high mortalities associated with changes in temperature. In addition, the obtained results could be extended to other species that share the same problem with low temperatures.
In my opinion, the work is worthy of being published in Animals. However, authors should review the following minor points before approval:
-Line 19: the word “occur” can be removed.
Thi word was removed as suggested
-Lines 14 and 29: LD50 and LD90 were higher, or lower? Please check.
It has been corrected
-Line 45: Fish inhabiting such habitats have developed
Corrected
-Figure 1a: Please indicate in the figure the temperature (18-20) at which the fish were kept during the growth trial.
Done.
-Line 100 and bellow: The authors mentioned that 500 fish were used. However, only 300 fish were used for both the Fish growth trial and the cold challenges. Please check.
Initially we acclimated 500 fish. However, we only used 300 randomly selected fish for experiments.
-Line 102 and Figure1a: 500L or 5,000L?
Fish were acclimated in 5000 L tanks, and growth trial was done in other tanks with a volume of 500 L
-Lines 134 and 135: please review this sentence. It is not very clear whether all the fish were pooled, or one of the two tanks was dedicated to each diet.
Sentence has been changed to clarify this (lines 146-147).
-Figure 2: please check the colours for the identification of the groups. The figure legend and figure caption do not match. In addition, in order to be consistent with Figure 1, the CTRL group should be presented in blue.
All figures have been changed in order to unify figures legend and caption.
-Line 300: capacity... increases
Corrected
-Lines 298-302: The authors mentioned that the requirements of PL in fish decrease with their development due to an increase in the capacity to endogenously synthesize PL with ageing. Said that the results show that fish fed the PL diet presented increased SGR in the first half of the growth trial, but not in the second part (Figure 2B). It might be thought, therefore, that the growth-promoting effects of PL supplementation will not be found in older animals. The authors should discuss this a bit deeper this section, and revise the use of “however” in line 301, as the observed results do not contradict the classical assumption about PL requirements.
Thanks to the reviewer for this observation. We think that differences in growth rates in the first half of the growth trial are difficult to be attributable to maturation or ageing. The animals used in the trial are juveniles (mean 12 gr) and in this species gut maturation occurs quite early during development. We think that these differences could be more associated with the role of PLs in energy provision and stress coping. This was now discussed in the MS.
-Lines 326-327: Considering that PL group grew more than CTRL animals, it is surprising that, as showed in Table 2, the wet weight at the death of PL-diet fed fish was lower than of the CTRL group. The authors neither mention nor discuss these results. Were the death fish of the PL group outliers for the body weight and SGR?
Fish for cold challenge were selected in a similar size to avoid that effect of weight on fish survival after the cold challenge. So, fish weights were normalized cold challenges and they should be used as a reference that challenged fish had not differences in growth and the differences in survival were associated with the dietary treatments. This aspect has been clarified in the text (lines 156-158).
-Line 363: Considering the presented results, that conclusion statement is only accurate for meagre under 30-35 grams (the first part of the growth trial). Please review it or indicate the limitations of the study and its derived conclusions.
The conclusion has been modified in accordance with modifications made in the first part of discussion.
Round 2
Reviewer 1 Report
The authors have not improved the quality of the MS and I regretfully reject it.
Kind regards